# Domesticity 'Behind Bars': Project by Rem Koolhaas/OMA for the Renovation of a Panopticon Prison in Arnhem

**Elena Martinez-Millana** [1,2,*] and **Andrés Cánovas Alcaraz** [1]

1   Departamento de Proyectos Arquitectónicos, Escuela Técnica Superior de Arquitectura,
    Universidad Politécnica de Madrid, 28040 Madrid, Spain; andres.canovas@upm.es
2   Department of Architecture, Faculty of Architecture and the Built Environment, Delft University of
    Technology, 2628 BL Delft, The Netherlands
*   Correspondence: elena.martinez.millana@upm.es

**Abstract:** This article focuses on the project for the renovation of a Panopticon prison in Arnhem, the Netherlands (1979–1980), designed by Rem Koolhaas/OMA. The analysis of its reception shows that, despite being well known, it has been little studied and discussed, and although it was not built, it had an impact on prison architecture. It seems appropriate to tackle it now because the *Koepelgevangenis* (dome prison) of Arnhem has gained current relevance due to plans for it to be turned into a hotel. The renovation project for the *Koepelgevangenis* explicitly shows the presence of Foucault's ideas on power and how these ideas exerted significant influence on the works carried out by Koolhaas. For Foucault, the Panopticon prison, such as the *Koepelgevangenis*, was the paradigmatic example of what he called the "disciplinary society". Domesticity "behind bars" suggests that prisons can also be understood as domestic spaces. Moreover, it could be said that for Koolhaas, this Panopticon prison was a social condenser or a hotel for voluntary or involuntary prisoners. As a prison or as a hotel, it can also be interpreted as Foucault's heterotopia, the intervention thus acquiring a new meaning which anticipated the future of this unique building.

**Keywords:** domesticity; sinister; panopticon; prison; heterotopia

## 1. Introduction

In April 1979, Rem Koolhaas (Office for Metropolitan Architecture) was commissioned to carry out the renovation and new construction of the *Huis van Bewaring* (detention centre), in the unique *Strafgevangenis* (penal prison), popularly known as *Koepelgevangenis* (dome prison), located in the city of Arnhem, in the Netherlands. Koolhaas' project was defined by the original report he presented to the authority in charge of the commission in March 1980, the *Rijksgebouwendienst* or *RGD* (The Government Buildings Agency), even if the actual study carried on for longer than that [1]. As Joost Meuwissen recently noted, unlike what happens in other of Koolhaas' works, this project, despite being well known, has hardly been studied or discussed. In a brief article titled "Self Portrait of a Society. Panopticon Prison, Arnhem", Meuwissen warned about this situation and its cause, stating that "the report (of the study) was so hermetic and apodictic that it was barely open for discussion" [2] (p. 16). Thus, this article aims to contribute to the investigation of the *Koepelgevangenis* or "dome prison" in Arnhem and offer a new point of view about the project by Koolhaas for the renovation of this building.

The hypothesis of this article arises as a result of a current discussion about the future of the *Koepel* and the changes regarding its use or occupation which have been taken place in recent years. *The Penitentiaire Inrichting De Berg* (Penitentiary Institution in Arnhem) closed in 2015 due to the decrease in prisoners in the Netherlands and their high maintenance costs [3]. Shortly after, 350 to

400 refugees were allocated there by decision of the *Centraal Orgaanopvang Asielzoekers* (Central Agency for the Reception of Asylum Seekers), as so were other empty prisons throughout the country. In 2018 the *Rijksvastgoedbedrijf* (The Central Government Buildings Agency) sold the complex and it is foreseen that a hotel will be opened there in 2021 [4]. Thus, the "dome prison" which has been a National Monument in the Netherlands since 2001, steers a new direction outside that of the *RGD* and its original use. Given this new turn of events, it is worth questioning: will the "dome prison" be able to get rid of the use for which it was conceived for in 1886?, Furthermore, if the project Koolhaas proposed in the first place had been constructed: How would it have transformed the domesticity of the prison? What possibilities would that domestic space offer "other" uses such as a hotel?

This article suggests that the study and analysis of the proposal Koolhaas carried out may discover a new type of relationship between the current situation and the future one for this unique building. As if this project had assumed—independently of the intentions of its authors—a new meaning, the hypothesis of this investigation has for its title "domesticity 'behind bars'". For "domesticity 'behind bars'", on the one hand, it has the consideration that prisons are also dwellings and must be treated as so. On the other hand, it can also be understood from this expression that there are certain aspects within a dwelling which we could call "sinister", just as Sigmund Freud defined it in *Das Unheimlich* (1919) [5] (pp. 335–376). According to Freud, the most "sinister" experience happens surrounding the domestic environment since it has to do with the intertwining between the idea of the *Heimlich* (what is characteristic to the house, and which is hidden, invisible to the eye) and the *Unheimlich* (what is not characteristic of the house and is shown even if it ought not to be visible to the eye) [6] (pp. 219–225). The renovation of the Panopticon prison of Arnhem acquires, from this point of view, today, a prophetic or visionary meaning of it.

## 2. Materials and Methods

This article is structured in two parts. Firstly, there is an introduction of the theoretical framework from which Koolhaas' project is analysed and, secondly, the analysis of the specific project, showing why this case study is of great interest to analyse the concept of domesticity "behind bars". Thus, the article is in line with those works such as the one by Hilde Heynen, who via the term "domesticity", studied the relationship between housing and modernity, showing its tensions and contradictions [7] (pp. 1–29). Moreover, the design of this investigation aims to achieve a "projective" analysis, which has resounded since the beginning of the XXI century and which authors such as Sarah Whiting and Robert Somol were the first to introduce [8] (pp. 72–77). In what is known as "projective" debate, the ideas that are put forward try to potentially reintegrate architectural practice and theory. As Lara Schrijver explains, the notion of a "projective" debate is a response to the need to look beyond material conditions and limitations which are present in the practice of architecture. According to Schrijver, " . . . it offered a line of demarcation, opening up the possibility to discuss the potential of architecture rather than its impotence" [9] (pp. 123–127). This is why a previous, solid theoretical framework has been built, which will then be integrated into the analysis of specific architectural practice.

In the first part, the peculiar nature of the building in which the intervention is to take place, the *Koepelgevangenis* in Arnhem (1886), is analysed, a penal prison which was conceived according to the Panopticon principle developed by Jeremy Bentham (1787). Thus, how the *Koepel* was carried out—almost a century after—and what the original bases of the Panopticon were is studied. After that, the issue that will be tackled is how this principle was received during the period when Koolhaas was commissioned with the project (1979), via the ideas of Michel Foucault and his discourse on power, which had a key role in the intellectual discussions of the 1970s. In the second part, the text deepens into the subject of up to what extent some aspects of the intellectual context aforementioned can be useful to study and debate—today—the proposal Koolhaas presented (1980). However, before showing the results, the existing relationship between the theoretical framework and the work which precedes it is put forward, as this is a relationship which has also been debated by other authors and which shows that this link between them goes beyond the intervention proposal for the *Koepel* (and this

means it has an added value in terms of its relevance). By means of this analysis, the aim is to make progress in line with other recent investigations which study domesticity, debating about its tensions and contradictions, beyond the traditional conception of a dwelling.

## 3. The Panopticism in Architecture: Theory, Evolution, Buildings

### 3.1. The Koepelgevangenis (Dome Prison) of Arnhem

The *Koepelgevangenis* was designed by the architect and engineer Johan Frederik Metzelaar (1818–1897), and it was built in the Dutch city of Arnhem between the years 1882 and 1886. Metzelaar was head of the first departmental building agency at the Ministry of Justice, created for the construction of prisons and court buildings, and these were his final years at the service of this institution [10] (pp. 15–29). Due to the introduction of the Penal Code in 1881, the minister A.E.J. Modderman (1838–1885) wanted Metzelaar to build as many prisons as were necessary since this code opted for a broader application of the cellular system. The *Koepelgevangenis* in Arnhem was one of the cellular penal prisons built for when this new code entered into force in 1886 [11] (pp. 9–14). Metzelaar designed the *Koepelgevangenis* for the cities of Arnhem, Breda and The Hague. However, he only managed to build the one in Arnhem (1882–1886) and the one in Breda (1883–1886). The *Koepel* of Arnhem was the first to be built and the one which served as an example for the others (Figure 1). The one in Haarlem (1899–1901) was built later by his son, W.C. Metzelaar. This design was different from the usual and a more spread out design for prisons compared to those that had been built until then both in the Netherlands and abroad.

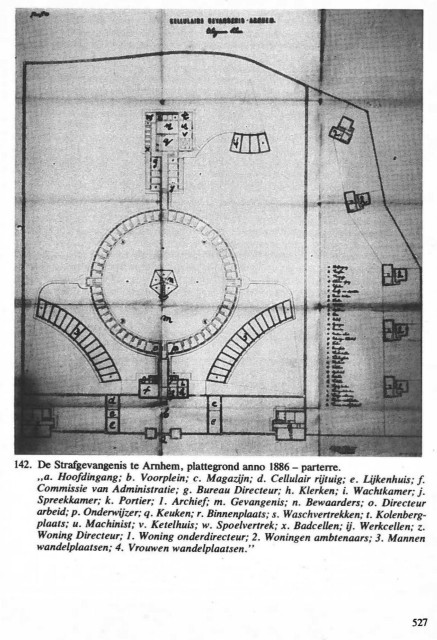

142. De Strafgevangenis te Arnhem, plattegrond anno 1886 – parterre.
,,a. Hoofdingang; b. Voorplein; c. Magazijn; d. Cellulair rijtuig; e. Lijkenhuis; f. Commissie van Administratie; g. Bureau Directeur; h. Klerken; i. Wachtkamer; j. Spreekkamer; k. Portier; l. Archief; m. Gevangenis; n. Bewaarders; o. Directeur arbeid; p. Onderwijzer; q. Keuken; r. Binnenplaats; s. Waschvertrekken; t. Kolenberg-plaats; u. Machinist; v. Ketelhuis; w. Spoelvertrek; x. Badcellen; ij. Werkcellen; z. Woning Directeur; 1. Woning onderdirecteur; 2. Woningen ambtenaars; 3. Mannen wandelplaatsen; 4. Vrouwen wandelplaatsen.''

527

**Figure 1.** The *Koepelgevangenis* or dome prison of Arnhem, Netherlands, by J.F. Metzelaar, 1882–1886.

Even though the ideas Bentham put forward had a great influence concerning the design of prisons, these buildings continue to be an exception. The *Koepelgevangenis* is one of the few examples of prisons built following the pure Panopticon principle Jeremy Bentham envisaged in 1787 [12] (p. 313). Marinus Albertus Petersen studied in depth the construction of the prisons designed by Metzelaar in *Gedetineerden onder dak* (1978). This doctoral thesis focuses on the history of the penitentiary system in the Netherlands from 1795. According to Petersen, to Metzelaar, the advantages of the "dome prisons" versus the "radial prisons" were clear: they were better in terms of isolating the interns and there was greater clarity and fewer construction costs. Among other reasons was that a church building was no longer necessary since Mass could be said from the pulpit located in the centre of the circle.

Both the minister and the regents from Arnhem and Breda were convinced by the proposal. However, the regents from The Hague strongly rejected the proposal stating clear objections in terms of the acoustics of the dome. Afterwards, in Arnhem and Breda, it was necessary to build separate churches since the prisoners could not understand the cleric from their cells, and this gave way to a great deal of criticism from the Lower Chamber [13] (p. 872).

The Panopticon architectural model consists of locating all the cells following the circular perimeter. Therefore, all the prisoners can be observed from a single surveillance point located in the centre of that circle. This distribution creates a large central courtyard which, in the design for the *Koepelgevangenis*, has a diameter of 52 m and is covered by an enormous dome with a height of 31 m [14] (p. 522–536) (Figure 2). The ring of cells is divided into four equal parts by four towers where a staircase made of stone connects the lower level with the other three levels. In addition, in the footbridges which give access to the cells, there are four metallic spiral staircases halfway between the towers. There is a total of 208 cells, slightly trapezoidal, with a width of 2.40 m on the door side and 2.80 m on the opposite side, where the window is. The length or depth of the cells is 4.00 m. The windows of the cells measure 1.20 × 0.70 m. The doors, which are made of pinewood, open towards the outside and are equipped with a peephole or a window with a lid. Apart from these, there are another two sets of fifteen cells outside the dome, a total of thirty outdoor cells, which are connected to it through pathways from the north and south tower.

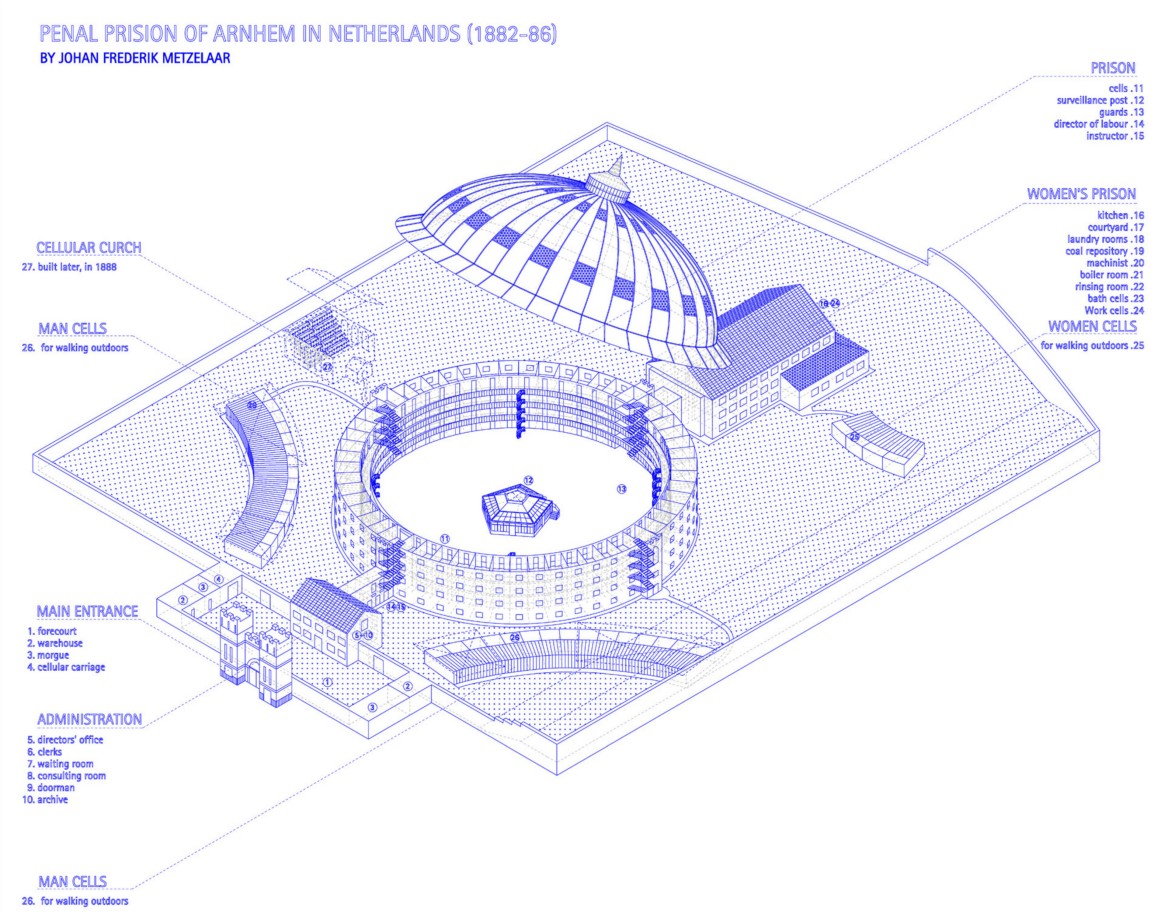

**Figure 2.** Drawing made for this analysis of the *Koepelgevangenis* in Arnhem, Netherlands.

Around the *Koepel* (the dome), there are other annex buildings which shape the complex of the prison. The one which holds the cells for women is connected to the main building via a passageway behind the western tower, which has also got four outdoor cells [14] (p. 530). In the street which gives access to the prison, the *Wilhelminastraat*, a monumental doorway, was erected as well as a series of

administrative and facility buildings [13] (p. 873). It is worth highlighting that the church was built a few years later (1888) due to the peculiar acoustics of the building, as previously mentioned. Thus, in the original plan one cannot find the church. The pulpit was located above the ceiling of the central surveillance post of the *Koepel*. According to Petersen, there were more drawbacks to this design: the difficulty the guards had to get closer to the cells without being seen by the prisoners—there was no inspection gallery to hide them—, and the impossibility of supervising the interior of the cells from the surveillance post—the walls of the cells were built of bricks. These "drawbacks" would be crucial in the study carried out by Koolhaas, who also made a point stating it had been strongly criticized from its origin as a "blatant luxury". It was common to hear people question if it was justified for prisoners to be accommodated in a building which resembled a hotel: "Wouldn't it promote crime instead of discouraging it?" [15] (p. 94).

### 3.2. The Revision of the Panopticon Principle

Jeremy Bentham (1748–1832), father of utilitarianism and famous legislator, was responsible for the idea of the Panopticon or House of Inspection in 1787, a century before the opening of the *Koepel* in 1886. Robin Evans, in his article titled "Bentham's Panopticon. An Incident in the Social History of Architecture", introduced the Panopticon in the architectural discourse of the 1970s [16] (p. 21–37). In this article, Evans explained that Bentham came up with it parting from a structure designed by Samuel Bentham, his younger brother, while he reorganized the properties of the Russian Prince Potemkin. This "proto-Panopticon" should have been a factory located in the city of Kritchev. However, the Turkish-Russian war took prominence, distracting Potemkin's attention from the local issues to the international ones. Thus, the project failed [17] (pp. 453–455). As Evans demonstrated, "Jeremy had joined Samuel in Russia and from there, in 1787, he wrote a series of contrived 'letters' (a common device for arranging descriptive material for publication at that time) setting out in excruciating detail his ideas for the design of institutions on the Panopticon plan" [16] (p. 21). According to Evans, the Penitentiary Panopticon was to contain about 460 prisoners in a rotunda of approximately 36.5 m diameter (120 feet) (Figure 3). Thus, it had more cells than the *Koepel* of Arnhem, 460 vs. 208 cells, and was smaller, 36.5 vs. 52.0 m in diameter.

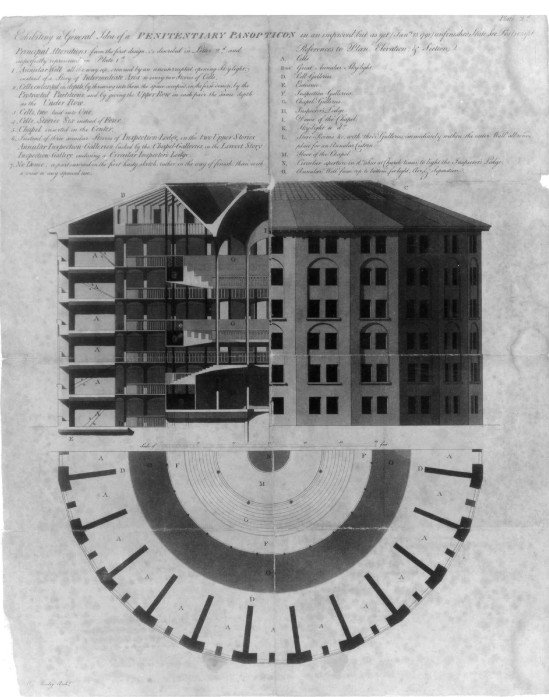

**Figure 3.** The Penitentiary Panopticon, by J. Bentham, 1787. This is the improved 1791 project drawn up by W. Reveley.

In the first lines of the first letter Bentham wrote regarding the Panopticon, he explains his scheme is "a way of obtaining power, power of mind over mind, in a quantity hitherto without example and that, to a degree equally without example, secured by whoever chooses to have it so, against abuse" [18] (p. 31). According to Evans, Bentham's term utilitarianism did not only imply the rigorous compliance of the functional requirements, but it also implied the incorporation of architecture to a sort of moral philosophy: architecture was supposedly an instrument to improve human beings, "as a catalytic agent inducing human goodness or reformation as part of a purely mechanical operation." Thus, Evans, in the article "Bentham's Panopticon. An Incident in the Social History of Architecture", presented the Panopticon as "the most significant monument to a forgotten creed that linked human betterment with architecture above all else" [16] (p. 21). This article was the origin of the subject he further developed in his doctoral thesis, presenting his dissertation five years later, in 1975: the architecture of prisons in the XVIII and XIX century, titled *Prison design, 1750-1842: a study of the relationship between functional architecture and penal ideology* [19].

The Panopticon prison model conceived by Bentham had a great influence not only in the field of the design of prisons of these centuries but also in the intellectual field of the second half of the 70s of the XX century. The person who was mainly responsible for this was the French philosopher Michel Foucault, who considered this model of prison a paradigmatic example of what he called "the disciplinary society". In *Surveiller et Punir. Naissance de la Prison* (1975), Foucault stated that the most important effect arisen by the Panopticon is that of "inducing the prisoner into a strong permanent and conscious belief that he is visible to the eye of who must control him, thus, guaranteeing the automatic functioning of power. Making surveillance permanent in its effect, even if it is not so in action. That the perfection of power may turn useless its actual execution; that this architectural apparatus may become the machine capable of creating and sustaining a relationship of power independently from the who exerts it; in sum, that the prisoners find themselves within a situation of power of which they, themselves, are the bearers" [20] (pp. 202–203).

According to Foucault, the Panopticon showed in some way the "diagram" of the disciplinary society: "The Panopticon must not be understood as a dreamlike building: it is the diagram of a mechanism of power referred back to its ideal form; its functioning, far from any obstacle, resistance or friction, can be very well represented as a pure architectural and optical system. It is, in fact, a figure of political technology which can be and must be detached of any type of specific use". Therefore, just as Bentham had planned it, Foucault established a relationship between the modern techniques of surveillance, present in the design of the Panopticon, with other architectural typologies apart from the prison, such as schools, factories, hospitals or military buildings, among others [20] (p. 207). This interpretation was closer to the Bentham definition, such as he had planned the Panopticon principle for initially, than to the narrower application that was made later only for prison architecture. All this largely explains why Foucault's analysis of Bentham's Panopticon had such impact on the architectural discourse of the second half of the 1970s.

The Panopticon was originally planned as a model for all types of institutions where the control of humans, or even animals, was considered important. Even if it tends to be associated only with penitentiary architecture, Bentham himself thought that it could be as useful for other purposes, "no matter how different, or even opposite", such as schools, hospitals, lazarettos, poor-plan buildings, houses of correction, lunatic asylums, orphanages, nurseries, institutions for the blind and deaf, homes for deserted young women, factories, and even a gigantic chicken coop [18] (p. 34). This list brings to mind the one Foucault made when he defined the concept of heterotopia, more explicitly spatial, in the lecture entitled "Des espaces autres" at the *Cercle des études architecturals* (1967) [21] (pp. 46–69). That term was coined before he published his study on prisons and panopticism, *Surveiller et Punir* (1975). The concept heterotopia was introduced and then immediately abandoned by Foucault; however, it had a great impact in the field of architecture. Both concepts became part of the architectural discourse during the 1970s, and both were introduced at the same time in the seminar "Il dispositivo Foucault" at the *Instituto Universitario di Architettura di Venezia* (1977) [22]. According to Michael K.

Hays, this was the perfect occasion for the theory of architecture to criticize the fundamental concept of Foucault's heterotopia together with the concept of Bentham's panopticism [23] (pp. 296–297).

In this review of the origin of the Panopticon principle and how it was received two centuries after, during the 1970s, it is possible to find that this concept was associated with the one of heterotopia, a concept which was already part of Koolhaas' discourse and which can be explicitly found in the dictionary of the book *S,M,L,XL* by OMA, Rem Koolhaas and Bruce Mau. Under the term "Animals" one can see the quote Foucault used when he first coined the term [24] (pp. xxii–xxiv). It is worth noting that a vast number of recent works rewrite the concept of heterotopia from the contemporary perspective, and therefore, this association becomes increasingly significant overall. It happens in "Heterotopia and the City", edited by Michiel Dehaene and Lieven De Cauter, where they replace it as a crucial concept in the current debate concerning the transformations carried out in both architecture and cities today [25] (pp. 3–9). Hilde Heynen also supports the idea that "pursuing the idea of heterotopia offers a productive strategy" to investigate the conditions of urban and social contemporary life [26] (pp. 311–323). Moreover, these and other authors, such as Christine Boyer in "The many mirrors of Foucault and their architectural reflections", have demonstrated that it is actually possible to associate these concepts with some of the first projects carried out by Koolhaas, showing that they are explicitly or potentially linked to both [27] (pp. 53–73). This is why the next step is to introduce a revision of these previous associations since they provide the key aspects to study the intervention proposal for the *Koepel* by Rem Koolhaas from this other point of view (Figure 4).

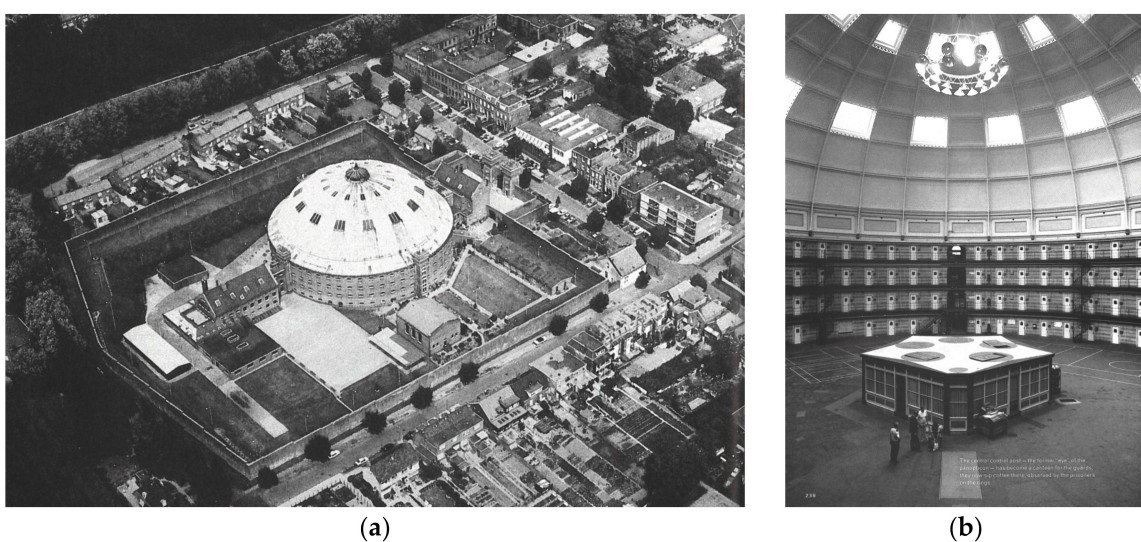

(**a**)　　　　　　　　　　　　　　　　　　　　　　(**b**)

**Figure 4.** Photographs of the *Koepelgevangenis* or dome prison for the study by R. Koolhaas/OMA. (**a**) Aerial view of the prison; (**b**) the central space of the dome, the surveillance post and the cells.

*3.3. The "Panopticism" in the Prior Works by Rem Koolhaas/OMA*

A year before starting to work on the project for the renovation of the *Koepelgevangenis*, *Delirious New York* (1978) was published [28]. This book was the culmination of a process of historic investigation regarding the American metropolis Koolhaas had carried out throughout the 1970s and which set the theoretical basis of his career as a project architect. It could be said that this investigation started off with two projects Koolhaas carried out in his training phase as an architect at the Architectural Association School in London: *The Berlin Wall as Architecture* (1971), and *Exodus, or the Voluntary Prisoners of Architecture* (1972) [29,30] (p. 212–232, p. 2–21). According to Koolhaas, the predominant architectural culture of those days had moved away from "real architecture", and he wanted to change that. He did so by designing a series of rather unconventional projects which served, in a certain way, as "intellectual Molotov cocktails" [31] (p. 263). Both projects gave faith to the original interest Koolhaas had in trying to understand what the real power of architecture was, "Exodus tried to go a

step further—a genuine unmasking of the real power and also genuine in what it was really about" [31] (p. 276). In this sense, both projects are closely linked to the intervention proposal for the Panopticon prison of Arnhem.

In "The Exodus Machine", Heynen and De Cauter study the genesis of the *Exodus* project, its content and context. They come up with a series of questions which they aim to give an answer to, among which two are key to understanding the context prior to the project for the prison of Arnhem: "Why this fascination with inclusion and enclosure? Why the sinister humour of the voluntary prisoners—'those strong enough to love it'?". They find the answer to these questions studying the genesis of the *Exodus* project, and it was Elia Zenghelis who gave them another key in order to do so: "the voluntary prisoners really did exist", referring to an article in the magazine *Time* or *Life* regarding American prisoners who preferred their life in prison rather than out of prison [31] (p. 263). That is why, in the genesis of *Exodus* one can find, in addition to the project of the Berlin Wall and the fortified city of skyscrapers in the desert of Zenghelis, the voluntary prisoners of the magazine *Time* or *Life* [31] (p. 274). The reference to the "voluntary prisoners" takes one back to the discourse on power by Foucault. However, in the genesis of *Exodus* it is not mentioned.

Nonetheless, in the interpretation Heynen and De Cauter issue of the project *Exodus*, they do explicitly associate it with Foucault's discourse, introducing the concept of heterotopia in their analysis. They explain that *Exodus* has assumed—independently from the intentions of its authors—"a new meaning": the project has been demonstrated to be prophetical, as some sort of "premonition of the rise of a capsular civilization": the gated communities, the enclaves, malls, theme parks, atrium hotels are all examples of a "capsular architecture" and a "hetero-topical urbanism" that, in *Exodus*, find a "legendary, conceptual prefiguration" [31] (p. 267). According to the authors, this anticipation is related to the obsession at the time with heterotopias, so they take into consideration this concept when analysing the *Exodus* project. As happens in heterotopias, *Exodus* is at the same time a camp and a theme park, and "it is this unlikely combination which lends Exodus its unmistakable visionary character" [32,33] (pp. 8–23, p. 723). This analysis of *Exodus*, where it is suggested that the project may be interpreted as a heterotopia is very much related to the hypothesis presented here, associating the *Koepelgevangenis* in Arnhem with this concept.

In the academic environment of the AA School, one can also find other keys to these first projects developed by Koolhaas. Roberto Gargiani studied this period in depth, and he relates these first projects to those projects and articles by two other students of the time, one of them being Robin Evans [34] (pp. 3–13). During that period and within that academic environment, Evans wrote several articles closely related to Koolhaas' discourse, such as the aforementioned "Bentham's Panopticon. An Incident in the Social History of Architecture" (1970) regarding the origin and history of the Panopticon, beyond its application in the architecture of prisons [16] (pp. 21–37). In "Towards Anarchitecture." (1970), Evans makes a point on the relationship between architecture and freedom, trying to clarify the direct effect of "things" on "human actions", this effect being beneficial or harmful, liberating or restraining [35] (pp. 58–69). He also wrote another article even more related to the subject: "The Rights of Retreat and the Rites of Exclusion: Notes Towards the Definition of Wall" (1971), where he studies the wall and he genealogically establishes a relationship between spaces of retreat as monasteries, and spaces of exclusion, such as prisons, which share the concept of life away from society. Even if in one case it is a voluntary decision, and in the others, it is not [36] (pp. 335–339).

After his studies at the AA School in London, Koolhaas moved to the United States to carry out his investigation work on New York. During this American period, Koolhaas spent a period at the University of Cornell (1972–73), where he got the chance to meet Michel Foucault [37] (pp. 348–377). Foucault gave one of his first talks in the United States in Ithaca in 1972. In a conversation with Beatriz Colomina, Koolhaas explained to her that he was able to meet Foucault thanks to the mediation of Hubert Damisch, a history of art professor and member at the time of the Department of Humanities at the University of Cornell. It is at that time that Koolhaas comes into contact with Foucault's ideas and his discourse on power. Foucault during that period was preparing his book *Surveiller et Punir.*

*Naissance de la Prison* (1975) [20] (pp. 197–229). Later on, the influence on Koolhaas of Foucault's ideas on power was studied by various authors [38] (pp. 18–21). The most significant analysis carried out was one by Hans Van Dijk, published in 1978, under the title of "Rem Koolhaas: de reïncarnatie van de moderne architectuur", where he outlined some of the main ideas of Koolhaas' projects in relation to this theoretical discourse. It is worth mentioning that it is possible to find the explicit reference OMA, Rem Koolhaas, and Bruce Mau make to the discourse on the "disciplinary society" by Foucault in the later book *S,M,L,XL*, where it appears to be quoted in different terms such as "Power" and "Exclusion" [39] (p. 384; p. 1052).

With regard to the article by Hans Van Dijk, published a year before Rem Koolhaas/OMA received the commission to carry out the study of the renovation of the *Koepelgevangenis*, in the section titled "het grid als manifest", Van Dijk associates the reference Koolhaas makes to "voluntary prisoners" with Jeremy Bentham's Panopticon principle. Van Dijk asserts that the grid that the functionalist avant-gardes envisioned as a "completely neutral organizational tool" is taken to the next level with Koolhaas, to an "architectural manifesto" [40] (p. 14). What is more, according to Van Dijk, in Koolhaas' projects, the addition of cells in a non-hierarchal manner substitutes the principle of Bentham's radial cells. He explains that, by eliminating the centre of power from the Panopticon, he shows it is no longer necessary since the inhabitants of those cells have become "voluntary prisoners" [40] (p. 15). The buildings Koolhaas takes as his models are hotels and social condensers, while Bentham considered factories, prisons, or hospitals, among others, as appropriate for his Panopticon principle. According to the explanation presented by Van Dijk, all these applications of the Panopticon are depictions of what Koolhaas loathes the most: "condescendence by means of architecture and "therapeutic" justifications" [40] (pp. 14–15). It does not seem coincidental that it was shortly after when he started carrying out the study for the renovation of the Panopticon prison in Arnhem.

### 3.4. The Project for the Renovation of the Panopticon Prison in Arnhem (1979–1980)

Foucault's ideas had a great influence on the intellectual discourse of the 1970s and consequently in architecture, particularly in Rem Koolhaas/OMA's work, and his study for the renovation of the *Koepelgevangenis* in Arnhem is the project where this influence became most explicitly clear. This influence can be seen in some of Koolhaas' statements about this project, such as the ones he made in October 1980 in a lecture at the *Technische Universiteit Delft*, where he explained this project: "Very few parts offer us such a faithful self-portrait of society as the prisons system does!" [41] (p. 95). This statement is not casual and is related to the critical conjecture which this article presents, domesticity "behind bars". The project and the explanation which accompanied it "disappeared" in the drawer of the Ministry of Justice. However, this proposal did manage to "set a change regarding penitentiary policies". Koolhaas demonstrated that the *Koepelgevangenis*, 100 years old, and with a "totalitarian aspect", offered greater opportunities for "more humane penitentiary policies" than the new ones built during the 1970s in the Netherlands [42] (p. 76). The focus of this section is on studying the renovation proposal, in order to analyse how the intervention transforms the existing domesticity at the *Koepelgevangenis* or the Panopticon prison in Arnhem. To that aim, the report presented to the *Rijksgebouwendienst* or *RGD* (The Government Buildings Agency) in March 1980 is studied and the extent to which this proposal is related to Foucault's discourse on power in the 1970s is analysed [1].

This study aims to go a step further than the recent work carried out by other authors, such as, for example, the one by Whitten Overby "A Multimedia Panopticon: Media, Translation, and History in OMA's S,M,L,XL and the Arnhem Prison (2015)". Overby studies the project for the Panopticon prison in Arnhem together with the later book *S,M,L,XL* (1995). Even if the study carried out by Overby has a different hypothesis and different aims, there are some similarities between Overby's study and the one presented here, which reinforce the hypothesis that has been put forward in this particular study. For example, the author places "Revision"—which is the title given to the project for the *Koepelgevangenis* in the book and which only contains a short version of the text presented to the *RGD* (1980)—as a third manifesto following the ones by Foucault (1975) and Bentham (1791),

"to detail the Panopticon as an emblem of contemporaneous space at large" [43,44] (p. 168, pp. 44–49). It is also worth making a reference to the recent work by Ingrid Böck, in her study *Six Canonical Projects by Rem Koolhaas* (2015), where she dedicates a chapter to the project *Exodus, or The Voluntary Prisoners of Architecture* (1972). More specifically, in "The Wall as a means of division, exclusion, and difference", the author puts forward a series of research questions linked to the *Exodus* project and which are intimately related with the ones this study poses: "How can planning relate to the creation of unprecedented events and new forms of social life? How do architectural decisions function as powerful means of order, division, and control? What are Koolhaas's strategies to present alternative ways of planning and decision-making?" [45] (p. 42). Moreover, Böck refers on several occasions to projects linked to this one, like the one which is analysed here, the Panopticon prison in Arnhem (Figure 5).

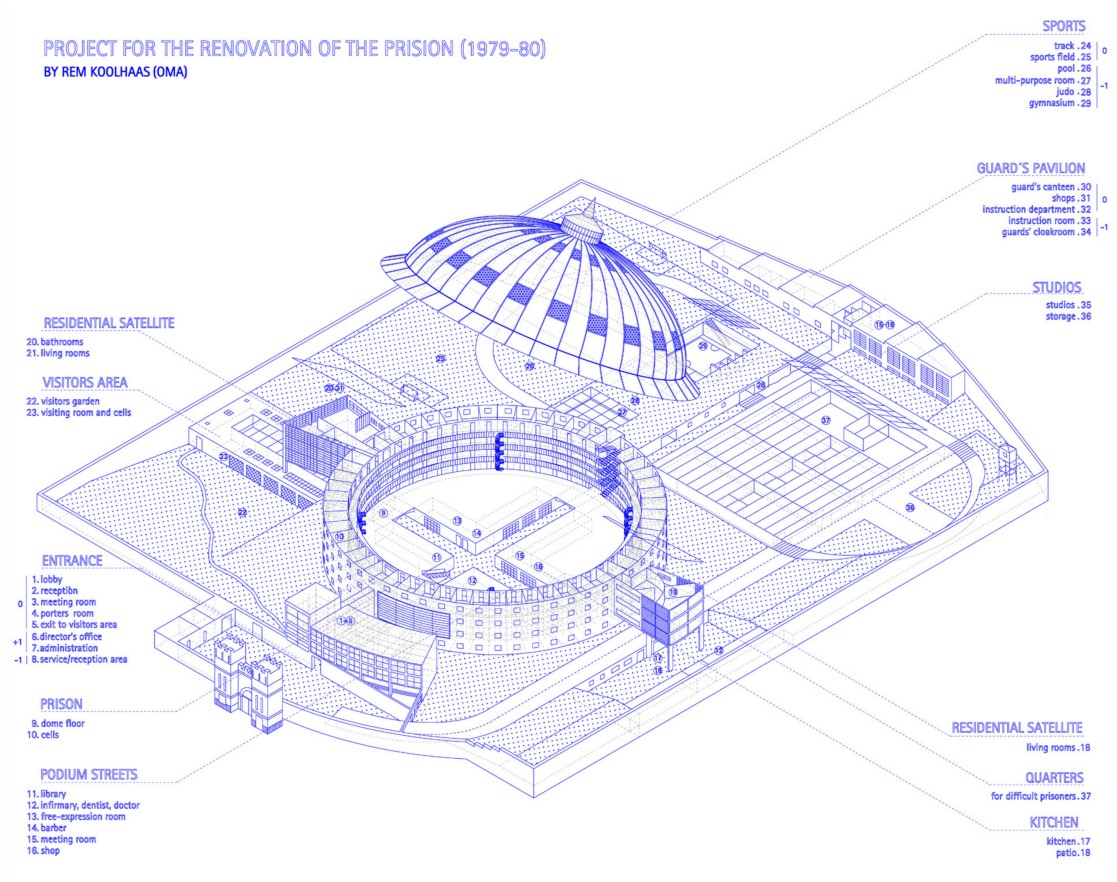

**Figure 5.** Drawing made for this analysis of the *Koepelgevangenis* renovation project in Arnhem, Netherlands (1979–1980).

### 3.4.1. The Client and the Commission

According to the requirements of the client, the building had to be equipped for it to be able to function "for at least another 50 years", and most importantly, it ought to "embody present-day insights into the treatment of prisoners ... " [46] (pp. 233–253). In order to study and discuss the proposal Koolhaas initially presented, it is necessary to outline "the origin" of the main propositions and principles which had been developed since he had been commissioned with the study of the renovation of the *Koepelgevangenis* on 12 April 1979. Koolhaas left proof of this issue at the beginning of the written description of the project, where he explained, among other things, what the point of view of those responsible for the commission and those supervising it was, throughout the study [42] (pp. 76–83). For example, in his first interview, W. G. Quis, the *Rijksbouwmeester* (Chief Government

Architect of the Netherlands) did not consider the "usability" of the *Koepel* to be strictly utilitarian; it also had to be examined from " . . . a semantic, symbolic, psychological, architectural, metaphorical point of view . . . ". He also wrote about the importance the supervisor of the commission had, the engineer B.H. Menke, from the Department of Projects 1; the renovation of the *Koepel* of Arnhem offered Menke the opportunity of reversing—or at least not repeating—"certain alienating aspects of recent buildings"; for example, Menke found favourable the direct contact between guards and prisoners, "they were tangibly 'on a boat' within the global space" [42] (p. 77).

The study was also supervised by Ir. F. Sevenhuysen, who, in line with the *Van Hattum* report, emphasized the importance of having a certain condition of "normality" within the prison, which meant that, "apart from the inevitable detention, the conditions inside the prison must differ as little as possible from those on the outside" [42] (p. 82). Koolhaas considered this a difficult issue to solve. In his own words, which somehow remind us of those ideas Foucault presented, he asked himself: "Is the 'normality' actually positive for the prisoner, or is it meant to make mankind outside the walls forget the embarrassing fact of the continuing existence of the prison?" [42] (pp. 80–81). Thus, in the renovation of the *Koepel*, a whole series of new facilities had to be implemented to be able to ensure the extension of the life cycle of the building. However, even more so, there was a requirement for the transformation of the general concept of the building, which somehow had to explicitly prove the change in penitentiary policies. If we bear in mind the influence of Foucault here, one can appreciate that the ambition of this project transcends the specific requirements of the commission, acquiring a great symbolic dimension. To provide a solution to this double set of requirements, Koolhaas developed a strategy which worked at different levels of significance. It could be said that Koolhaas took the opportunity to continue exploring the "real power of architecture", symbolically dismantling a building which was already far too full of ideological meaning.

### 3.4.2. The Genesis of the Project

In order to analyse the genesis of the proposal that Koolhaas presented to the *RGD* in March 1980, the study covers not only the original report but the presentation he made on a postdoctoral course titled *Lessen in architectuur,* which took place in October 1980 at *TU Delft*. During this lecture, Koolhaas exclusively explained the project for the renovation of the *Koepelgevangenis* he had presented only a few months before then. Because of both the content and the format itself, it is a document of great interest, which complements the original report and enables a much better analysis of the genesis of the project. This presentation was published in the *Syllabus van de leergang,* together with the images he showed in his lecture and the questions posed by other guests at the course [41] (pp. 92–105). Koolhaas started off his presentation explaining his interpretation of the skyscraper, the Downtown Athletic Club, as he had done two years before in *Delirious New York.* For Koolhaas, the interesting thing about that building was the way in which "a boring rectangular floor plan creates human activity chains", "new scenarios for humanity". Koolhaas was interested in manipulating, designing or organizing programs so that "new activities, never seen before, meaningful or meaningless" could take place [47] (p. 21). In this association between the project for the *Koepel* in Arnhem and the formula for skyscrapers, it was Meuwissen who made a point, strongly asserting that had the programme been uncomplicated, the design could have brought about the paradigm shift in architecture that was established a few years later by the "horizontal skyscraper" for *Parc de la Villette* in Paris [2] (p. 19).

Next, Koolhaas explained the nature of the *Koepelgevangenis* in Arnhem, introducing the concept of the Panopticon principle. In the presentation, he offered a more complete definition than the one he had given on other occasions, using very meaningful images. Koolhaas explained that Bentham had not only designed the Panopticon for prisons, but also for hospitals, factories, and other institutions, "where, for the first time, it was imperative to get involved with the new and modern masses". It was an "abstract architectural formula" which "acquired an eternal bleak meaning and for the first time was seared permanently in the history of architecture when it came across the idea that it could become the ideal shape of penitentiary architecture" [15] (p. 94). It must be said that Koolhaas,

to explain this, used one of the pages of Evans' article "Bentham's Panopticon: An Incident in the Social History of Architecture" (1971). See the coincidence of the drawings of the prisons in the following documents [16,47] (p. 29, p.22) (Figure 6). Even if he did not cite him, it is evident by the coincidence between the drawings Koolhaas presented and the ones Evans had formerly selected from different chapters—and even the appendix—of the book *Remarks from the Form and Construction of Prisons* (1826). Evans selected and compared the following four cases which appear throughout the book: (A) The County Gaol at Ipswich, 1790; (B) The Borough Gaol at Liverpool, also by Blackburn, 1779; (C) The House of Correction at Kirkdale, near Liverpool, 1829; (D) The Female Prison at Lancaster Castle, 1821 [48] (p. 18, p. 25, p. 26, p. 69). This is the reason why, on this occasion, the association is more than explicit, Koolhaas showing an in-depth knowledge of the origin and the meaning of the Panopticon principle.

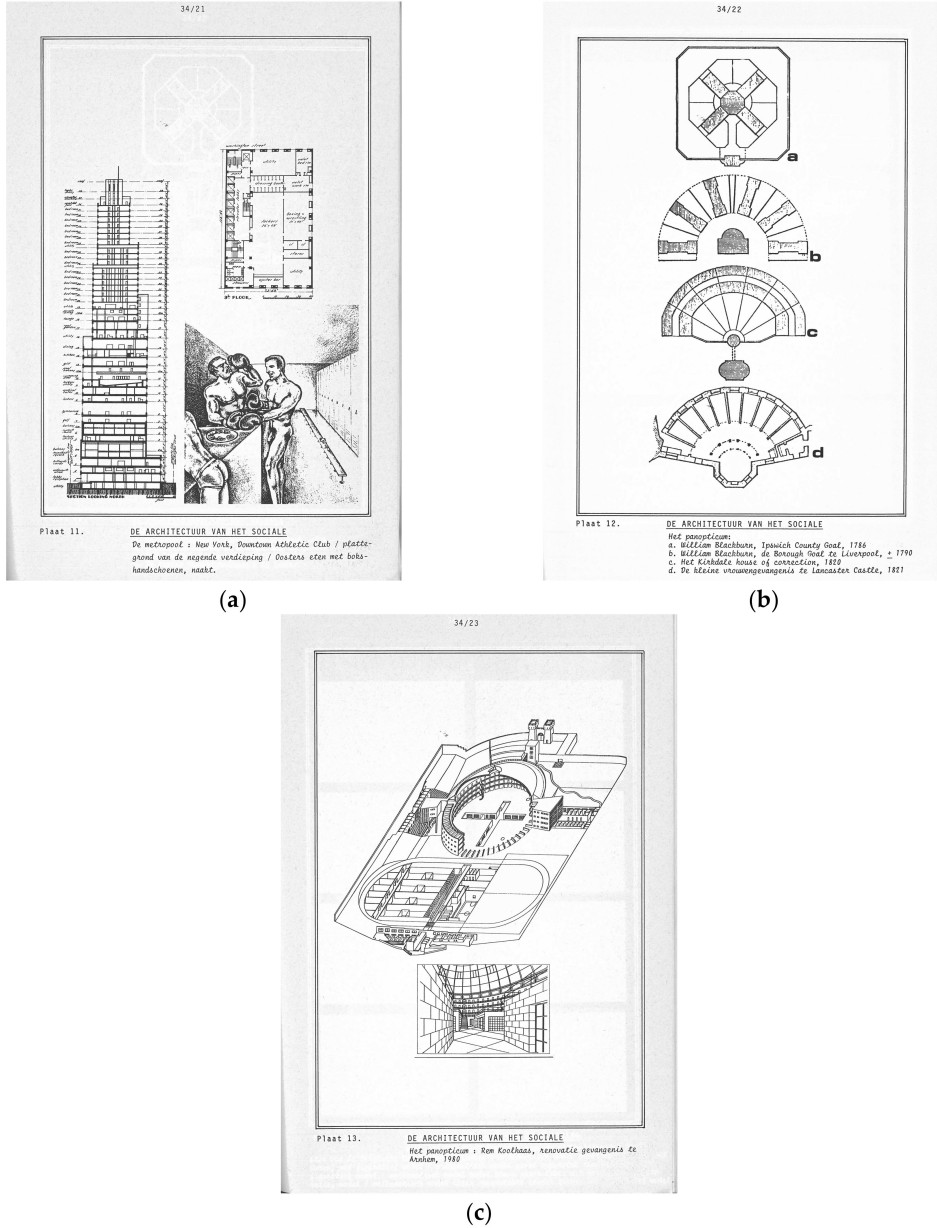

(**a**)                                   (**b**)

(**c**)

**Figure 6.** Images of the lecture *Een ontwerp voor de gevangenis te Arnhem* (A design for the prison in Arnhem) by R. Koolhaas, 1980, at *TU Delft*. (**a**) Plaat XI. "The metropolis"; (**b**) Plaat XII. "The Panopticon"; (**c**) Plaat XIII. "The Panopticon: Rem Koolhaas, prison renovation in Arnhem, 1980".

### 3.4.3. Developments of the Penitentiary System

The changes in points of view regarding the penitentiary system occupied a vast extension of the notes on the project. This also happened in the case of *Lessen in architectuur,* where he explained it even more extensively. The Panopticon prison or *Koepelgevangenis* in Arnhem had been conceived in 1882 according to what in those days was the "progressive ideal" of solitary confinement. According to what Koolhaas explains "both in terms of luxury and quality of the accommodation as in terms of the ideology behind it, it was a progressive and idealistic building". Soon after, as the XX century advanced, solitary confinement became unacceptable. Those prisoners who had been isolated had gone mad because of solitude [15] (p. 94). This is why the principles of solitary confinement were transformed little by little, turning into a broader programme which included work, sport, and leisure activities. A series of buildings with the "emergency" collective installations were added surrounding the "dome prison" [41] (p. 95). Besides these extensions, in 1958, there was a much more radical condemnation of the *Koepel* prison expressed by the Jacobs committee which got to the point of even suggesting its entire demolition and its substitution for new buildings. This committee started to think about how a "modern prison" should be, and for the first time, it established that it should not be organized in "one large mass" but via "smaller groups", that is, the "pavilion prison". With this, according to Koolhaas, the ambiguity which arose was that of "the hypocrisy of prisons being disguised as normal buildings" [41] (p. 96).

In the revision Koolhaas carried out of the different developments of the penitentiary system, he also analysed the newly built prisons in Amsterdam and Maastricht, which incarnated "the most enlightened ideas" of the 1960s. For Koolhaas, the Panopticon prison in Arnhem demonstrated being more "flexible", while modern architecture was based on a "deterministic coincidence between form and program". The aim had stopped being the "moral improvement" of the prisoner as the Panopticon principle suggested but "a literal inventory of all the details of daily life." According to Koolhaas, the "flexibility" in Arnhem was not an "exhaustive anticipation of all the possible changes" but the capacity of making possible the "different and even opposite interpretations and uses" [46] (pp. 239–240). The *Penitentiaire Inrichting Over-Amstel* (Penitentiary Institution Over-Amstel) in Amsterdam, commonly known as *Bijlmerbajes*, built between the years 1972 and 1978, already had many detractors—society in general and specialists too. Even before its opening, many were the protests which took place [49]. In *Bijlmerbajes* the floor plan of each tower shows a collective space with a community installation and several cells surrounding it. The creation of different groups in *Bijlmer* is given by the distribution of each plane or level of the tower, in "families" of prisoners. According to Koolhaas, the new prisons were synonymous with "excessive control" for different reasons: "By removing any sense of a collective, subdivision has reinforced feelings of isolation; the relationship with the guards has become mediated through electronic devices; the therapeutic pretension of the "family" unit has eroded the previous honesty of the guard-prisoner polarity" [46] (p. 240). The paradox is that this prison did not differ much from the surrounding dwellings; it did not even have bars to it.

### 3.4.4. The Renovation of the *Koepelgevangenis* as a "Revision"

As had already happened in the *Exodus* project, the strategy for the *Koepelgevangenis* arose largely from the analysis carried out regarding the behaviour of prisoners [31] (p. 263). According to the explanations given by Koolhaas in the notes relating to the project, at the moment when the situation was analysed, which in the *Koepel* was in 1979, it could be strongly asserted that those "current ideas" to which the supervisors assigned to the commission referred to were along the lines of the following: "present-day insights had already spontaneously changed—in fact, drastically reversed—the performance of the building" ( . . . ) "the former centre of power—the 'eye' of the Panopticon—had been converted into a canteen for the guards: the former observers are now themselves observed by the prisoners, who are no longer kept locked in their cells at all times but could circulate freely on the rings and have access to the ground floor. Originally envisioned as empty, the entire interior is now

often as busy as the Milan Galleria" [46] (p. 237). This actualization of domesticity proved that the *Koepelgevangenis*, intact for almost 100 years, had once again been able to assume the changes in the ideas regarding penal justice. In response, Koolhaas offers an architectural solution which seems to try and give continuity to that process of reversion—in relation to its domesticity—which had already spontaneously begun.

The *Koepelgevangenis* in Arnhem was a great opportunity for Koolhaas to investigate what he called "renovation as a revision", according to which, the "revision" was only possible where there once was a "vision". The *Koepel* met this condition because, within it, an "unusually direct relationship" occurred between an abstract principle, the Panopticon by Bentham and its architectural organization. Koolhaas considered that modern penitentiary architecture should consist of a "prospective archaeology", in the sense that the ideological layers that start arising show "how the ideas have progressed" from the initial starting point. Thus, the renovation of the *Koepel* was projected as a layer of "revision", "a new layer of 'civilization' or 'enlightenment' on top of the existing one" [42] (pp. 79–80). In this sense, in his study, Koolhaas aimed to verify if, apart from the spontaneous revisions that the building had been subject to, it was possible to carry out a second revision, which meant a "programmatic and ideological change" according to "current ideas", preserving the building itself. Koolhaas set the focus especially on two main properties the existing building had: the first one, the interior space itself in the context of a prison and, the second one, its symbolic clarity against the confusion which had been created with the development of those new prisons—such as the one of *Bijlmerbajes*—, which did not differentiate the building holding the prison from its surrounding residential buildings [42] (pp. 80–81).

### 3.4.5. The Domesticity in the "Public Domain"

The key concept of the proposal, just as Koolhaas describes it in the notes which accompany the project, is the "decoupling" between the old and the new. The most recognizable strategy of the intervention, that which can synthesize the attitude adopted for the entire proposal, is the intersection of two streets excavated in the floor of the large central space, which perpendicularly cross in the centre of the Panopticon: "The streets and the new collective facilities create a new scenario, a new ideological base on top of which the dismantled Panopticon is presented as a historic relic" [42] (p. 81). In this way, Koolhaas responded, with a combination of both pragmatism and irony which characterizes his proposals, to the two main requirements set by his clients. On the one hand, the central space was conveyed with a certain urban ambiguous nature, with a set of streets where there were shops and other facilities, projecting the metaphor of the passage of the XIX century, whose most clear example is the *Galleria Vittorio Emanuele* in Milan. On the other hand, the emptiness which is the result of the intersection of those two streets made it impossible to locate this as the surveillance post since "its intersection forever eliminated the "eye" of the panopticon" [42] (pp. 81–83). This will to "eliminate the centre", to "eliminate the eye" of the Panopticon is expressed with great clarity by the well-known image (photogram) of *Un Chien Andalou*, (Luis Buñuel and Salvador Dalí 1929), which appears at the beginning of the chapter dedicated to this project in *S,L,M,XL* [46] (p. 233, p. 235) (Figure 7). Koolhaas used two photograms from this film—both before and during the cutting of the eye, an eye that for Koolhaas represented the "eye" of Bentham's Panopticon—to emphasize the main strategy of the intervention proposal: the cutting of the "eye" of the Panopticon prison in Arnhem.

Through the introduction of these two streets, what was created at the same time was an "exterior world", transforming the dome exclusively into a "home". These underground streets connected the interior space with the exterior one, blurring the limits between one and the other and problematising at the same time the relationship between indoor and outdoor. In the words of the author, "The same relative freedom that now exists in the dome is extended across the two streets. In this way, essential contrasts that define life outside—such as indoors/outdoors, home/work, house/street—are re-established inside the prison" [50] (p. 60). This conception of the project put an emphasis on the condition of "normality", which was the core of all the new ideas on accommodation for prisoners. In the streets, that same level of freedom of movement prevailed, but also, that same

form of surveillance. With the intervention, all the enclosure of the prison became more organized, and, therefore, "easier to control than with the disorganized addition of emergency buildings" [42] (p. 81–83). These streets gave access to all the new amenities of the "public domain" within the prison. According to the description Koolhaas provides in his notes on the project, the *Zuiderstraat* leads to the visitor's centre, whose façade opens towards an inclined garden. The prisoners see the visitors exiting the waiting room via a path; the *Noorderstraat* gives access to an excavated courtyard where the kitchens, the medical department, and a special area for difficult prisoners are located; the *Westerstraat* leads to the more active functional area shaped by four workshops, sports equipment, and a room dedicated to cinema/theatre/religious events. The ceiling of the workshops is partially equipped for outdoor works such as gardening. Apart from the views towards the street, each workshop has a courtyard with plants. The stretch of street located in the sport's area shapes a swimming pool. The football pitch and the running track are accessible from the changing rooms located in the gym.

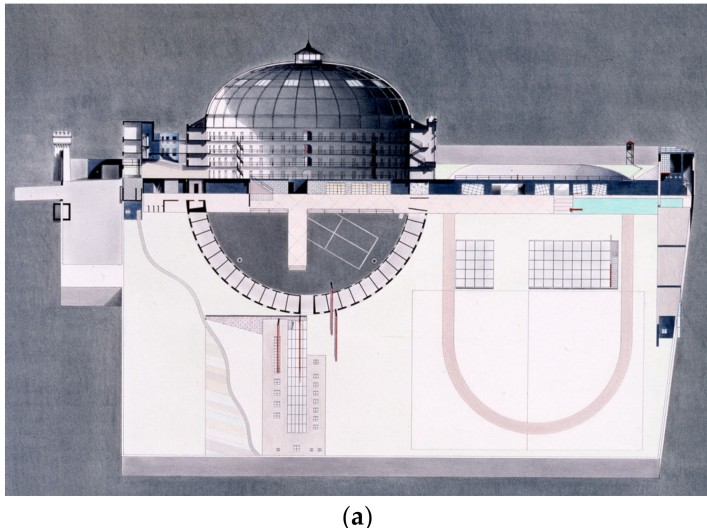
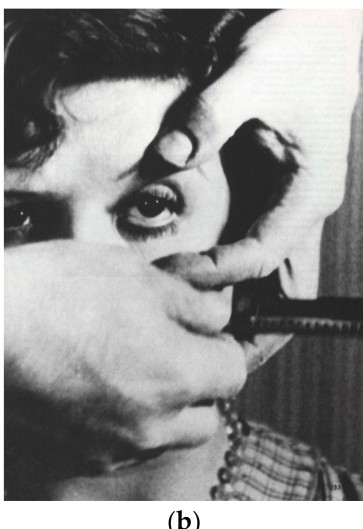

(**a**)                 (**b**)

**Figure 7.** The intervention proposal for the *Koepelgevangenis* in Arnhem. (**a**) The project: interior perspective drawing, by R. Koolhaas/OMA, 1979-80; (**b**) a photogram of *Un Chien Andalou*, L. Buñuel and S. Dalí, 1929. Koolhaas compared this with the way in which the intersection of the new streets eliminates "the eye" of the Panopticon.

### 3.4.6. Paradoxes of Domesticity "Behind Bars"

The cross of the two new perpendicular streets shaped in the centre of the *Koepel* gave way to various interpretations. Stanislaus Von Moos, in "From the orphanage to the dome prison. History as 'ready-made'", associated this composition with the *Prouns* of Lissitzky, comparing one of the axonometric drawings with the *Proun R.V.N.2* (1923) [51] (p. 59). Another quite different focus on this was the one by Mark Adang or Koos Bosma, whose perspectives are linked to the study of penitentiary architecture in the Netherlands. According to Adang, the justification of the cross as a symbolic dismantlement of the *Koepel* was "gratuitous and insufficient, because in that point there was never such amount of power" [52] (pp. 6–26). Actually, as Petersen suggested—and Koolhaas strongly affirmed—the construction of the *Koepel* from its origin had brought with it a series of "drawbacks" concerning the surveillance of the prisoners, which meant that the centre did not exist as such in the end [53] (p. 45). In this sense, Bosma described as a "pathetic sentence" Koolhaas' argument to justify the cross, and explained that he "created too many negative associations with it" [54] (p. 90). It was even said that it could be a "religious" infiltration [41] (p. 102). Adang searched for the cause of the mistaken assessment and suggested that the proposal was one according more to the projects Koolhaas did than with the contemporary ideas on penitentiary architecture. Proof of this is that in this proposal one can find "almost literally" the hypothesis Hans Van Dijk had developed in 1978 regarding the

architecture in *Delirious New York,* where he established a link between Koolhaas and Bentham [52] (p. 26).

In this sense, it could be suggested that, in the *Koepel*, the cross was part of a pattern of imaginary squares, such as the one underlying the city of Manhattan, and that in the projects by Koolhaas it was located as the axis of the non-hierarchical pattern against the principle of Bentham's radial cells. Thus, the axis of that grid in the interior of the *Koepel*, presented as streets or "passageways", dismantled its domesticity, eliminating the function with which the *Koepel* was associated. The *Koepel* was the materialization of an abstract principle associated with its original architectural organization, which transcended the prison use. That is why Koolhaas suggested the renovation of the *Koepel* as a "revision" of its original principle, updating its domesticity. This "revision", as had already happened with his prior projects, led to the discourse on power by Foucault. In this case, it would be in a more explicit way since Foucault considered the Panopticon prison as a paradigmatic example of what he named "the disciplinary society", and as Koolhaas expressed, very few parts of society offer us such a faithful self-portrait as its prisons system [41] (p. 95). The renovation shaped this thought, "decoupling" the old building and the new one; opposing and juxtaposing the scheme, the diagram shaped that abstract principle in the XX century against the one which crystallized in the XIX century, thus disassembling the past. In the words of Koolhaas: "After the intervention, the dome represents the dismantled past, its former centre crossed out, resting on a podium of modernity, which is concerned only with improving the prisoners' conditions" [46] (p. 247).

### 3.4.7. Another Domesticity in the *Koepel* "Dwelling"

Another important strategy of the proposal is the intervention in the *Koepel* or the "dome"; all the necessary facilities for it to work as a dwelling—living rooms, dining rooms and bathrooms— were all located in two satellite constructions. These were directly connected with the rings of cells but located in a sort of wedge shape in the exterior. This way, according to Koolhaas' explanations, the interior space of the dome would be preserved intact, and these new facilities placed in the exterior of the *Koepel* would be the proof that there had been an actual revision carried out [41] (pp. 100–101). Koolhaas put the focus on the key role of these satellite constructions in the creation of groups, comparing them to the way in which this was carried out in new prisons such as *Bijlmer* in Amsterdam, where the creation of groups was determined by the configuration of the building in "families". Thus, in the *Koepel*, the prisoners could distribute themselves in groups of about twenty-four members, and they could be even less numerous thanks to the subdivision created within each of the satellite constructions. Moreover, the vertical communication between the different cell rings made it possible for these groups to be formed by prisoners who did not necessarily live in the same ring (floor). According to Koolhaas, this was an extremely flexible regime, where the prisoner could be part of different groups, at different moments and for different reasons, "the group which is created each time is not an architectural piece of fiction but a momentary social configuration which can suffer endless permutations" [42] (p. 83). The individual cells of the *Koepel*, which had made it possible to have solitary confinement—now considered something negative—in this revision meant it made it possible for the prisoner to have freedom of choice.

In this way, the proposal offered the prisoners two types of collective spaces according to the different contrasts: such as interior/exterior, home/workplace, and dwelling/street. Apart from the collective space of the "outer world", which they could access via the new streets, there were other collective spaces as part of the "inner world" of the *Koepel*, in the "dwelling". As Mark Adang had explained in "*Gevangenisbouw in Nederland . . .* " in the history of the construction of prisons there is a great influence of both the penitentiary and the political points of view. Due to the fact that these are very varied, the prisons are the only buildings which can be labelled as "Hilton Hotels" and as "extremely sophisticated instruments of torture" [52] (p. 6). This ambiguity had chased the *Koepel* since its origins in a paradigmatic way. The proposal put through by Koolhaas showed the real nature of the Panopticon and the *Koepel*, as a Panopticon and also as a heterotopia. The *Koepel* is much more than a

prison. Thus, the project transformed the concept of domesticity in such a way that it freed the prison from the function it had already been assigned to have. The *Koepel,* as if it were a hotel, accommodated this community of voluntary or involuntary prisoners. For the administration, Koolhaas opted for the same solution. It was located in a third satellite construction in substitution of the old one, facing the main gate. The satellite of the administration was part of the same geometry or system. Therefore, it had arisen as a new ring projected from the centre: "ultimately, one can imagine an endless number of rings." The guards had an area reserved for them resulting from the four "exploded cells", a sort of common room which did not occupy a privileged position within the system.

*3.5. Prisons without Prisoners: The Architecture of Prisons When These Are No Longer Necessary*

In the introduction, when describing the hypothesis of the research, an important emphasis was placed on the growing interest which has arisen regarding the *Koepelgevangenis* in Arnhem. In the Netherlands, many prisons are empty due to the important drop in prisoners during the last decade. Among these are the three *Koepelgevangenis* built in the Netherlands, which were declared National Monuments in 2001. These three prisons have been maintained despite them having been closed, first the one in Arnhem in 2015 and then the other two in 2016, the one in Breda and the one in Haarlem. These prisons, together with the other empty prisons around the country, were temporarily used to give accommodation to refugees. Since then, they have been the subject of study with the aim of evaluating their possible reconversion into new uses such as hotels and dwellings. It is worth mentioning that, even if there has already been an attempt to define how some of these could be reused, an example of this being the *Koepelgevangenis* in Arnhem, which will be turned into a hotel, others on the contrary have already been demolished. In 2019, the prison Koolhaas studied and used as an example to put forward a strong criticism regarding the developments of penitentiary architecture during that period, was demolished (3.4.7. Another domesticity in the *Koepel* "dwelling"), the *Penitentiaire Inrichting Over-Amstel* (Penitentiary Institution Over-Amstel) in Amsterdam, commonly known as *Bijlmerbajes* (1972-78). This demolition holds even greater significance when analysing the study Koolhaas carried out on the penitentiary developments (3.4.3. Developments of the penitentiary system). With regard to the obsolescence of prisons he stated that: "it has become impossible to build a prison that is not, at the moment of its completion, out-of-date" [46] (p. 241).

Against this background, it would seem reasonable to ask oneself: "Would it be possible to take some of the aspects of the project for the *Koepelgevangenis* to carry out the reconversion of other typologies of prisons?" Even more, beyond other "dome prisons" or "Panopticon prisons" as for example a modern prison like *Bijlmerbajes* (1972–1978), one could also wonder: "in what sense would this be able to happen?" It is evident that it is not possible to make a direct transfer of the aspects in question to just any typology, since the key of the intervention is in the elimination of the eye of the Panopticon, which is the emblem of the "disciplinary society" which Foucault puts forward. However, it is no less true that if the Panopticon can show the face of the "disciplinary society", that is also so because it is a prison. One must highlight once again the example of the *Bijlmerbajes* prison, since Koolhaas noted the similarities between the modern prisons and the dwellings via this project and, moreover, inversely, the similarities between the dwellings and the prisons. This project led to many protests; it seemed intolerable for the society of the time that there were no significant differences between prisons and dwellings. As Jeffrey Kipnis noted in "Recent Koolhaas", "An architect may reasonably strive for a congruent event structure in a prison or a hospital, but such extreme congruence would be intolerable in a house" [55] (p. 30). Here, paradoxically, the opposite happened. Another key of the project is the observation of the Panopticon prison. When Koolhaas visited it, he was able to give faith to the fact that the elimination of the Panopticon eye was already happening, thus, the proposal was actually giving continuity to what was already happening there (3.4.4. The renovation of the *Koepelgevangenis* as a "revision"). In this sense, these strategies could in fact be taken from this prison to others.

Therefore, "What is it that distinguishes the renovation proposal by Rem Koolhaas/OMA for the Panopticon Prison in Arnhem from other projects regarding the renovation of prisons?" It can be asserted that the answer lies in the critical nature which characterises it. One could also wonder: "How does it acquire that critical dimension?" In this case, the answer can be found in the fact that this proposal by Koolhaas eliminates the difference between the conception of the house and the prison, between the inhabitant and the prisoner. The project does not treat the prisoner as such, nor does it treat the prison as a prison either. The fact that the architect does not make such a distinction, the fact that he eliminates this differentiation, gives way to a project of critical intervention because, as Koolhaas himself manifested: "very few parts of society offer us such a faithful self-portrait". For Koolhaas, as Kipnis explains, architecture is only capable of engendering provisional freedoms in a specific situation: "undermining select patterns of regulation and authority". Thus, it can be said that this project is a paradigmatic example of this attitude since, "he goes to some lengths to demonstrate that tangible, liberating experiences supported by architecture can be engendered in restrictive and totalitarian contexts" [55] (p. 27). In this sense, this article proves that the renovation of the Panopticon prison was an opportunity for Koolhaas, since he had the chance to make a criticism with regard to the "disciplinary society" put forward by Foucault, which suggests that prisons and dwellings can be included in the same category. Moreover, what was originally a criticism of penitentiary architecture has resulted, at present, in a positive contribution, as if that critical character of the proposal deep down could be considered a pragmatism which has turned to be prophetic. In Koolhaas' words: "Changes in regime and ideology are more powerful than the most radical architecture" [46] (p. 239).

## 4. Conclusions

This research has aimed to show how the project by Rem Koolhaas/OMA suggested transforming the domesticity of the *Koepelgevangenis* and, thus, what possibilities the *Koepel* could have offered beyond its original use if the project had actually been built. The hypothesis of the article which has been developed was placed bearing in mind the recent drift and future of this unique building, which will shortly be turned into a hotel. Thus, the critical hypothesis questions if this "dome prison" or Panopticon prison, which for Foucault is none other than the emblem of what he called the "disciplinary society", could in fact free itself from the use it was conceived for. Today, apart from the *Koepelgevangenis* in Arnhem, there are many other prisons in the Netherlands which are to be reconverted giving them new uses. That is the reason why, ultimately, this article has tried to identify which strategies of this proposal could be useful for the reconversion of other typologies of prisons beyond the "dome prisons". In the first place, to what extent has the Panopticon principle crystalized in the *Koepelgevangenis* at the end of the XIX century been proved, and why did it not spread across the Netherlands. The design and the construction of the "dome prison" had a series of inconveniences which led to it being unsuccessful. Among them, it has been highlighted how the guards were in extremely dangerous positions. Therefore, the *Koepelgevangenis* received strong criticism from its very start as a "blatant luxury". Secondly, what has also been proved is what the acceptance of this abstract principle was and of the building itself almost a century after its opening, in relation with Koolhaas' discourse and his study for the renovation of the prison. The study and interpretation which Koolhaas carried out of the *Koepelgevangenis* shows an influence of the ideas put forward by Foucault and his discourse on power over his work. However, even though this association goes beyond this specific project, it is in this one where his influence was evident in a more explicit manner.

This issue has been proved in the analysis carried out in this project, since Koolhaas eliminates the difference between the conception of the dwelling and the prison, between the inhabitant and the prisoner. The project does not treat the prisoner as such, nor does it treat the prison as one. This can be explained via two arguments which are mutually reinforced. The first one being the design of the proposal itself, whose key points are the creation of a public podium, "cancelling the original 'eye' of the Panopticon", and the fact that he incorporates in the *Koepel* an ambiguous urban quality in a way that there is no distinction between the interior and the exterior world. Secondly, the project

introduces a series of facilities into the ring of the cells which equip the historic relic so that it works as a "dwelling". The other key argument focusses on the criticism he made of the developments of penitentiary architecture. Koolhaas noted the similarities between the prisons being built during that period and modern dwellings, and this, paradoxically, seemed intolerable to the society of the time, thus questioning the limits of what was normally considered a dwelling. The article has demonstrated that this project was an opportunity for Koolhaas to literally interpret this Panopticon prison as a social condenser or a hotel for the society of "voluntary prisoners". What is more, it is worth mentioning that the project has assumed, independently from the intentions of its authors, a new meaning. The *Koepelgevangenis*, stripped or not of its original use, as a prison or as a hotel, can also be interpreted as a heterotopia according to Foucault, where Koolhaas' intervention acquires a prophetic or visionary meaning since the intervention was going to transform the domesticity of the prison offering it exactly this possibility. This transformation of the domesticity of the Panopticon prison in Arnhem, "behind bars", unmasks the tensions and contradictions which are inherent to it—the sinister dimension inherent to domesticity—because prisons are also dwellings, and inversely, dwellings can also be considered prisons.

**Author Contributions:** Conceptualization, E.M.-M. and A.C.A.; methodology, E.M.-M.; investigation, E.M.-M.; writing—original draft preparation, E.M.-M.; writing—review and editing, E.M.-M. and A.C.A.; supervision, A.C.A.; funding acquisition, A.C.A. and E.M.-M. All authors have read and agreed to the published version of the manuscript.

**Funding:** This research was funded by the EU EUROPEAN SOCIAL FUND and COMUNIDAD DE MADRID, grant number PEJD-2017-PRE-HUM-4149, and UNIVERSIDAD POLITÉCNICA DE MADRID CONSEJO SOCIAL Fellowship, and later EU ERASMUS+ Traineeship Grant.

**Acknowledgments:** The authors wish to acknowledge Dick Van Gameren, Dean of the Faculty of Architecture and the Built Environment at the Delft University of Technology, for his supervision and generosity. Regarding technical drawing support, the authors wish to acknowledge Teresa Moreno Blasco, who has made the drawings together with author E.M.M.

**Conflicts of Interest:** The authors declare no conflict of interest. The funders had no role in the design of the study; in the collection, analyses, or interpretation of data; in the writing of the manuscript, or in the decision to publish the results.

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
