# Peer review of "Domesticity ‘Behind Bars’: Project by Rem Koolhaas/OMA for the Renovation of a Panopticon Prison in Arnhem"

_buildings, doi:10.3390/buildings10070117_

Round 1

Reviewer 1 Report

This paper describes a quite known project of Rem Koolhaas: the refurbishment of Panopticon prison, located in the city of Arnhem. The prison was built in 1886, by W.C. Metzelaar.

This paper would demonstrate that the study and analysis of the Koolhaas proposal carried out a new type of relationship between the current situation (an abandoned prison) and the future one. The authors try to demonstrate that Koepel could remain, thanks to Koolhaas's visionary project, a prison or becaming a social condenser or a hotel.

Finally the authors study the contemporary mean of "domesticity" and the value of Koolhaas proposal.

The article is full of references, but which intends to demonstrate a predetermined idea: the innovative and visionary charge of the Koolhaas proposal.

some suggestions

3. the title results and discussion sounds too much "scientific" for a paper of design theory. I suggest The Panopticism in architecture: theory, evolution, buildings.

in line 341 I don't understand what is Annexe 1.

Reviewer 2 Report

General comments:
This article focuses on the project for the renovation of a Panopticon prison designed by Rem Koolhaas/OMA, offering a new point of view about the project for the renovation of the building.
The authors stress the importance of the term "domesticity 'behind bars'" taking into account that prisons are also dwellings and must be treated as so.
The point of view of other authors is questioned and correlated to the project of Rem Koolhaas/OMA, specifically: the Koepelgevangenis project in Arnhem, Netherlands.
The work is well structured with fluent English.
No critical issues emerge.
Some notes:
The captions in Figure 2 and 5 are difficult to read. Please improve the resolution.
Figure 3 and 6 should be enlarged to better understand the structure and arrangement of the cells.
Considering the large amount of data, the conclusions should be more incisive.

Reviewer 3 Report

The paper presents an interesting reading of the project for the renovation of the Panopticon prison in Arnhem, designed by Rem Koolhaas/OMA. Even if the project is quite well known, commented and published, as well as theories and applications of panopticon to prisons, the critical reading presented by the author/s is original and well described. The methodological approach is correct, the state of the art well documented and commented, as well as the results convincing and well supported.

Overall, the paper is good and well written. Some references are missing and, in my opinion, should be commented in the text. The references are:

  • Whitten Overby (2015) A Multimedia Panopticon: Media, Translation, and History in OMA's S,M,L,XL and the Arnhem Prison, Journal of Architectural Education, 69:2, 167-177, DOI: 10.1080/10464883.2015.1063396 (which has many analogies with the paper).
  • Ingrid Böck (2015). Six Canonical Projects by Rem Koolhaas. Essays on the History of Ideas. Jovis: Berlin, (see in particular the chapter 1).

Reviewer 4 Report

The projects described in the essay do make a contribution to the history of the work, but overall, a more critical enquiry into Evan’s claims, and into Foucault’s position is necessary. A general interpretation of an annular prison through the lens of Bentham is not revealing of anything new. A more rigorous and critical analysis of the concepts is necessary; the claims that Bentham, Foucault and Evans made have all been thoroughly critiqued in the literature, and accepting their ideas at face value is of little value. One could ask a few questions immediately : what did Foucault get wrong about disciplinary societies? If the panopticon is merely a diagram, what significant differences emerge when the diagram is built? In a related question : why are prison's not designed like panopticons? Is it because they don't work, because they are excessive cruel ? Because they place the guards in extraordinarily dangerous positions? Because the prisoners do not internalise the governance structures, but rather always see it as a externalised object ? etc?

The conclusions made at the end of the essay reflect the absence of a sufficiently ambitious aim. The statements made about the OMA project are so general that they could be made about almost any renovation project. Given Foucault’s and Koolhaas’ principles, it seems that all architectural projects would “represent Koolhaas’ society of “voluntary prisoners”’  or involuntary ones; and all projects can be interpreted as heterotopias if the community occupying the heterotopia is sufficiently narrowly defined. The terms of reference the essay engages in are too broad to generate any meaningful conclusion. 

The author should look outside architecture for advances in critical understandings of utilitarianism, prison design and disciplinary systems.

A few minor points:

p.3, / 106: 1883-86?

p.4 / 150: This meaning of this line is not clear: “the difficulty the guards had, in order to get closer to one of the cells passing unnoticed,” 

p.5 / 190: Grammar needs correcting : “The main responsible for this was…” 

p.9 / 337:“offered greater opportunities for “more humane penitentiary policies”  Curious to know what opportunities exactly arose here.

p.12/498, p.14 / 552 - The references to artists are too thinly explained and appear as merely comments about superficial similarity. Either a deeper engagement with these subjects or omit. 

Round 2

Reviewer 4 Report

The main problem is with the narrow focus of the research project that has been written about. Though the author claims that they have identified a unique document which contributes to the value of the research, the significance of this document is not clear.

Some biographical moments have been identified but their import is absent. Koolhaas encountering Foucault does not change the way we see the work. The connections between Bentham and Foucault and Koolhaas are explicit in Koolhaas' own book S,M,L,XL. 

As mentioned in the previous review, the author would benefit from looking beyond the narrow confines of architectural writing when examining prisons, discipline, Foucault etc. An extended period of research and rewriting will really improve the project begun here.

Round 3

Reviewer 4 Report

The essay is much improved with the additional sections.

The author should not draw much import from the marginalia of S,M,L,XL, as these were prepared by co-author Bruce Mau, not Koolhaas. 
